



# Comprehensive data set of in-situ hydraulic stimulation experiments for geothermal purposes at the Äspö Hard Rock Laboratory (Sweden)

Arno Zang[1,2], Peter Niemz[3,1], Sebastian von Specht[4], Günter Zimmermann[1], Claus Milkereit[1], Katrin Plenkers[5], and Gerd Klee[6]

[1]Helmholtz Centre Potsdam GFZ German Research Centre for Geosciences, Potsdam, Germany
[2]Institute of Earth Sciences, University of Potsdam, Germany
[3]University of Utah Seismograph Stations, Salt Lake City, UT, United States
[4]Institute of Mathematics, University of Potsdam, Germany
[5]GmuG mbH, Bad Nauheim, Germany
[6]SolExperts GmbH, Bochum, Germany

**Correspondence:** Arno Zang (zang@gfz-potsdam.de)

**Abstract.** In this article, a high-resolution acoustic emission sensor, accelerometer and broadband seismometer array data set is made available and described in detail from in-situ experiments performed at Äspö Hard Rock Laboratory in May and June 2015. The main goal of the hydraulic stimulation tests in a horizontal borehole at 410 m depth in naturally fractured granitic rock mass is to demonstrate the technical feasibility to generate multi-stage heat exchangers in a controlled way superior to former massive stimulations applied in enhanced geothermal projects. A set of six, sub-parallel hydraulic fractures is propagated from an injection borehole drilled parallel to minimum horizontal in-situ stress, and monitored by an extensive complementary sensor array implemented in three inclined monitoring boreholes and the near-by tunnel system. Three different fluid-injection protocols are tested: constant water injection, progressive cyclic injection, and cyclic injection with a hydraulic hammer operating at 5 Hz frequency to stimulate a crystalline rock volume of size 30×30×30 m at depth. We collected geological data from core and borehole logs, fracture inspection data from impression packer, acoustic emission hypocenter tracking and tilt data, as well as quantified the permeability enhancement process. The data and interpretation provided through this publication is an important step both, in upscaling laboratory tests, and downscaling field tests in granitic rock in the framework of enhanced geothermal system research.

## 1 Introduction

Climate change on Earth poses great challenges to the transition from energy production based on fossil-fuels to low-carbon renewables. Among renewable energy sources, geothermal energy provides a local solution for district heating and electricity supply. As such it has the potential to provide safe and clean energy for growing urban areas worldwide, although geothermal settings are also located in remote areas, e.g. The Geysers (USA), Coso (USA) or Cooper Basin (Australia). Conventionally, geothermal energy is extracted from deep rock formations that are naturally fractured and faulted. Permeable fractures and



faults can serve as fluid pathways, thereby improving fluid production from and injection to a reservoir. However, faults are also at risk to abruptly slip, resulting in seismic events caused by geothermal energy operations (Buijze et al., 2019; Zang et al., 2014; Majer et al., 2012). The risk of such induced seismicity is the major factor opposing the widespread development of geothermal energy. Injection-induced geothermal seismic risk must be addressed (Giardini, 2009) and new methods are required to make this environmentally friendly energy source available to the wider community (e.g. in urban areas) by mitigating the risk of larger induced seismic events (Bommer, 2022; Grünthal, 2014). From a seismic risk assessment perspective, the fatigue concept described and discussed in this study aims at reducing seismic risk by reducing the seismic hazard, not by changing vulnerability and exposure. This is insofar different from natural seismicity risk, as the hazard of naturally occurring earthquakes is considered immutable.

After the massive stimulation of a granitic rock mass under the city of Basel, Switzerland, in 2006 (Häring et al., 2008), apart from traffic light systems to control induced seismicity (Bommer et al., 2006; Baisch et al., 2019) also the idea of multistage fracturing in EGS wells evolved (Meier et al., 2015). The core idea of multistage fracturing is adopted from shale gas production perforated completion design (parallel and isolated fractures), which is applied to economically produce gas from horizontal wells. When adapted to an EGS, multistage fracturing will economically produce heat by increasing the size of heat exchangers in a controlled way; one heat exchanger per perforation (Glauser et al., 2013; Fleckenstein et al., 2022). Planned in 2014 and executed in 2015, the Äspö underground experiment described in this study, was a mine-scale test campaign in naturally fractured granitic rock demonstrating the safe development of sub-parallel, zonal-isolated multi-fractures as potential heat exchangers for geothermal purposes (Zang and Stephansson, 2019). After completion of the Äspö experiment, similar and more sophisticated underground campaigns followed, e.g. the 2016+ stimulation at Grimsel Test Site, Switzerland (Amann et al., 2018), the 2018+ Homestake Mine underground tests at Saniford Facility in North Dakota, USA (Kneafsey et al., 2018), the Stimtec experiment in Reiche Zeche (Boese et al., 2022; Plenkers et al., 2023) and a planned larger-scale fracture and fault stimulation experiment in Aar granite at Bedretto Underground Laboratory in Switzerland (Ma et al., 2022).

There is a substantial lack of in-situ observations of acoustic, geologic, and hydraulic properties of rock mass at the intermediate scale in the literature (Amann et al., 2018). The importance of such meso-experiments is twofold: first, mine-scale tests allow to upscale laboratory tests, and to downscale field tests in naturally fractured rock. This is an important point to better understand the physical mechanisms behind hydraulic fracture growth. Does the fluid-injection scheme at laboratory scale impact the stimulation process and seismic response of the rock mass in a similar way as tests at the mine and field scale? If not, what are the driving factors of fluid-induced fracture onset and growth at different scales? Is it the pressure distribution in the fracture, the release of natural proppants from the fracture walls, or the leak-off of fluid into the surrounding rock which dictates hydraulic fracturing versus hydraulic shearing? How can we properly precondition an enhanced geothermal system (EGS) reservoir for hydro-shearing? What are the mechanisms of fluid-injection-induced seismicity at different scales? The second aspect of meso-experiments is of economic nature: the cost of mine-scale tests is far less and more controllable than field-scale borehole operations. Furthermore, evidence of geothermal concepts (multi-stage heat exchangers) on mine-scale are more reliable than small-scale laboratory tests of granite cores or cubes. At different scales, fractures and faults may exhibit similar sensitivity to permeability-stress, but can also result in very different stress and flow relationships. There is a definite

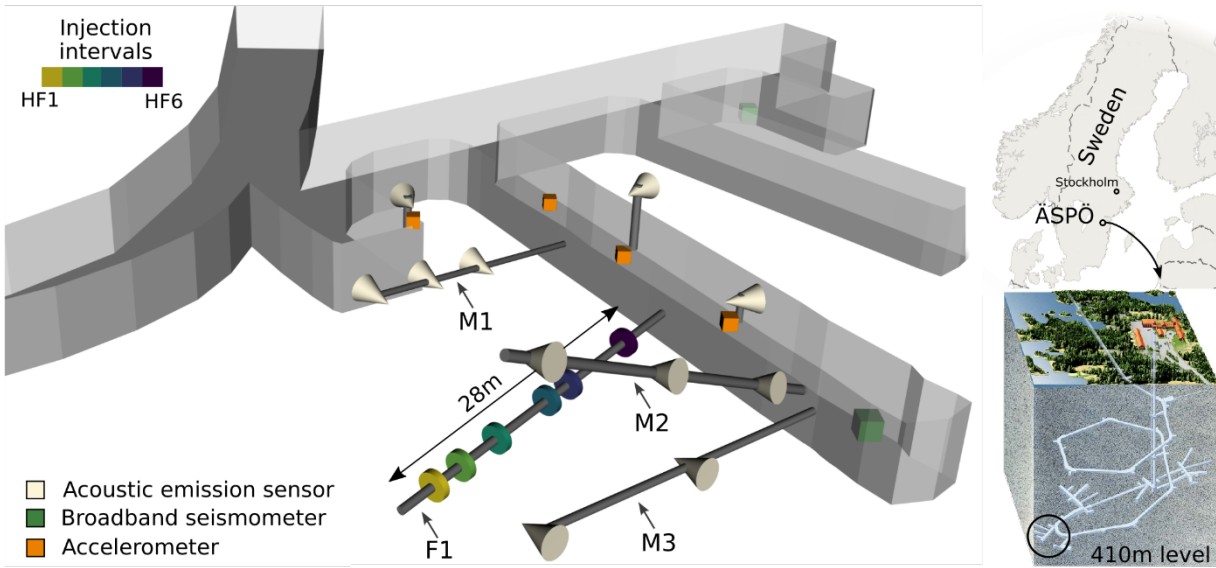

**Figure 1.** Location (top right) and overview (bottom right) of Äspö Hard Rock Laboratory with the test site at 410 m depth (left). Access tunnels are shown in lighter grey. The test borehole F1 with six hydraulic tests indicated by color and the monitoring boreholes M1-M3 with sensors (cones and cubes) are described in the following sections.

need to investigate the fracture and fault response to mechanical stress and fluid injection in similar rock types at different scales (Zang et al., 2021).

In the framework of a decameter-scale in situ stimulation experiment, a naturally fractured granitic rock mass was selected at Äspö Hard Rock Laboratory (HRL), on Äspö Island, Sweden, at 410 m depth. The site was chosen for its well-known setting in terms of geologic parameters, hydraulic properties and in-situ stress magnitudes (Zang et al., 2017). The Äspö HRL is owned by the Swedish Nuclear Fuel and Waste Management CO (SKB) and has been in operation since 1995 for nuclear waste disposal research. In close cooperation with the municipality of Oskarshamn, SKB has supported the formation of Nova FoU (R&D) with the aim to use the facilities at Äspö HRL for research and development in other fields. Here, we present a comprehensive data set of the geothermal project Nova 54-14-1 carried out at Äspö HRL. The site hosts 1.8 Ga old intrusive rocks ranging from granite, syenite, diorite to gabbro. These rock types can be assumed as representatives for a deep crystalline basement, which is generally the target rock of deep geothermal energy systems.

In this work, we present a comprehensive data set consisting of geologic data from four ∼30 m long boreholes (one horizontal stimulation borehole and three inclined monitoring boreholes) and seismic waveform data recorded by acoustic emission sensors, accelerometers and broadband seismometers. The seismic data set was obtained during six hydraulic fracturing tests with three different fluid-injection protocols (conventional constant flow rate, cyclic progressive water injection, and cyclic plus hydraulic hammer fatigue testing). The seismic and hydraulic rock response was recorded and subsequent data analysis provides insights into evolution of permeability and fracture development. The data accompanying this article includes waveforms

recorded by acoustic emission sensors and accelerometers (operating in triggered and continuous mode) as well as broadband seismometers in the near-field of six hydraulic tests each with multi-stage hydraulic fracturing (total of 29 stages). Further-more, the data set covers seismic and hydraulic properties of the host rock during water injection and back flow. After the tests,

the fractured rock mass was characterized using impression packer results, combined with analyses of acoustic emission and broadband sensor tilt signals observed. Subsequent data compilations include acoustic emission catalogues, seismic magni-tude estimates among others. Laboratory experiments on cores from the stimulation borehole complement the comprehensive mine-scale data presentation.

## 2 Methods, set-up of the experiment and tests

In May 2015, the Äspö experiment started by diamond drilling fresh boreholes into naturally fractured rock at 410 m depth. The rock mass was characterized using geologic, geophysical and hydraulic methods. Dry and wet cores from the injection borehole (diameter 102 mm) and three upwards inclined monitoring boreholes (diameter 76 mm) were analyzed. An optical borehole televiewer (BIPS tool) was used to map the injection borehole wall before and after the stimulation treatment. Geophysical techniques include a borehole tomography with an ultrasonic emitter sensor inserted in the injection borehole before and after

propagating six hydraulic fractures. The hydraulic techniques include a Lugeon test at borehole scale, and the computation of rock permeability with time. Cores from the injection borehole (diameter 86 mm) taken from the fluid-injection borehole at respective packer intervals of the six hydraulic stimulations were investigated in the laboratory.

### 2.1 Geology

#### 2.1.1 Test site and geologic background

The island of Äspö and its surroundings is located in the Transscandinavian Igneous Belt (TIB) of the svecofennian orogen which forms the core of the Fennoscandian shield in northern Europe (Stanfors et al., 1999). The bedrock in the TIB is dominated by well preserved, approximately 1.8 Ga old intrusive rocks varying in composition between granite, syenite, diorite and gabbro. The most prominent ductile structure on Äspö intersects the island in an NE-SW direction (deformation zone NE2, e.g. Ask, 2006). Subsequently, the rock mass is subjected to repeated phases of brittle deformations under varying regional

stress regimes and followed by reactivation along earlier generated structures. With few exceptions the deformation zones in Äspö HRL are of a brittle type, complex, and involve several reactivation events. The complexity of the fracture system at the test site is illustrated by the presented drill cores from the central part of the hydraulic testing borehole and selected borehole images (see section 2.1.2).

The underground test site is located in the Äspö extension area 2011-2012, at depth level of 410 m below surface (Stenberg,

2015). Based on the geological and hydrological description of Stenberg (2015), a borehole geometry was conceptualized with one injection and several monitoring boreholes. Figure 1 (left) provides an overview of the test site and its surroundings. The experiment was set up in access tunnel TASN from which four long boreholes were drilled (Fig. 1 left). The central



borehole (F1) serves as injection borehole and was drilled to a total length of 28.40 m subparallel to the orientation of the minimum horizontal compressive stress, inclined downward by $4°$. The monitoring boreholes (M1-M3) were drilled with
upward inclination. Owing to the presence of a hydrological conductor producing up to 75 mLs$^{-1}$ outflow of water, sensor installation in this way optimized the array signal sensitivity and coverage.

Geospatial data at Äspö HRL is provided in a site-specific coordinate system, ASPO96. All infrastructure coordinates (tunnels, boreholes etc.) are given in this coordinate system and we adopted its usage, due to the lack of access to any other geospatial reference system underground. Conversion from $(X, Y)$-coordinates of ASPO96 (in m) to the universal transverse
Mercator (UTM) projection on the WGS84 reference ellipsoid with Easting and Northing $(E, N)$ in UTM zone 33 (in m) is as follows:

$$\begin{pmatrix} E \\ N \end{pmatrix} = \begin{pmatrix} \cos\alpha & \sin\alpha \\ -\sin\alpha & \cos\alpha \end{pmatrix} \begin{pmatrix} X - X_0 \\ Y - Y_0 \end{pmatrix} + \begin{pmatrix} E_0 \\ N_0 \end{pmatrix}, \tag{1}$$

with $\alpha$ = -11.823°, $X_0$=2368.614, $Y_0$=7404.361, $E_0$=599947, $N_0$=6366968.

### 2.1.2  Analysis of drill cores and boreholes

The extracted cores from the boreholes were photographed in dry and wet conditions (Fig. 2a,b). We consider mechanical discontinuities as fractures regardless whether they are secondary mineralized or not. This allows to determine the fracture density per meter in each borehole. Combining the core logs with borehole televiewer logs (Fig. 2c) yields a detailed description of fracture density for the injection borehole. Borehole images are obtained with a borehole camera observation tool (BIPS, Döse et al., 2008).

On average, we observe four fractures per meter in the 86-mm-diameter drill core of the horizontal injection borehole. Our selection of test sections for hydraulic fracturing is based on the borehole images We identified four different rock types in the injection well. Close to the access tunnel TASN (section 0-6 m) the injection borehole consists of Äspö diorite and fine grained granite (fgG). The following sections (6-17 m and 17-28 m, at the end of the borehole) consist of fine-grained diorite-gabbro (fgDG) and Ävrö granodiorite (AG), respectively. Fig. 2c shows some of the geology along the borehole in the range 24-26 m
and the fractures intersecting the drill cores.

### 2.2  Borehole geophysical methods

### 2.2.1  Ultrasonic pulse transmission tests

An ultrasonic pulse transmission test was performed before the fracturing experiments on 02. June 2015. For the measurement ultrasonic transmitter GMuG-Tr40 (main frequency range 1 kHz to 50 kHz) was installed inside fracturing borehole F1 using a
pneumatic system to couple the source to the borehole side wall. The ultrasonic transmitter was installed inside the fracturing borehole at positions with 1m spacing between 1.36 m and 24.36 m borehole depth. The sensor was facing upwards (12 o'clock orientation). At 6.36 m, at 12.36 m, at 18.36 m and at 24.36 m borehole depth the sensor was installed in addition in 3 o'clock and 9 o'clock orientation, when seen from the access tunnel (TASN) wall. At each position 1000 pulse signals are emitted





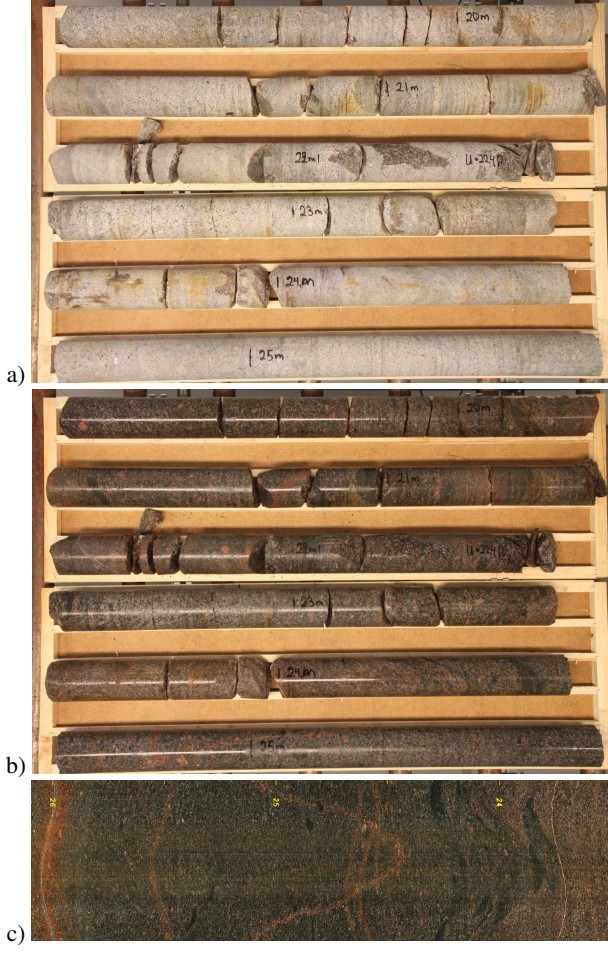

**Figure 2.** Fracture inspection from drill cores (section from 19 m - 26 m) : a) dry, b) wet. c) Image from borehole televiewer BIPS tool before the experiment in the section where test HF1 was conducted (24 - 26 m) .

and automatically stacked by the recording system. The stacked waveform was stored (window length 32 ms). The ultrasonic
pulse transmission data was used to estimate P-velocity ($5810 \pm 120$ ms$^{-1}$) and S-wave velocity ($3400 \pm 200$ ms$^{-1}$) within the
experimental rock volume.

### 2.2.2  Hammer seismics

Before and after the experiment, hammer seismics with a sledge hammer were conducted along the walls every 2 m of the
access tunnel TASN and the neighboring tunnels TASA and TAS02. The hammer strikes are among the strongest recorded
signals and are of much higher amplitude on all sensors. Exact localization and timing—usually recorded during hammer
seismics experiments—of the hammer points have not been determined, as the idea to perform hammer seismics originated on



site. Nonetheless, this data subset can be of value, if a joint inversion of hypocenter determination and seismic tomography is performed.

## 2.3 Hydraulic methods

This section gives an overview of the three different injection schemes. We describe the hydraulic well test analysis and the Lugeon test for the determination of the rock permeability and discuss results from the impression packer to determine hydraulic fracture orientations.

### 2.3.1 Fluid-injection schemes

In the experiment, three different hydraulic testing procedures were applied. The first test design is a conventional procedure
with constant flow rate (Fig. 3a). For the second testing procedure, the flow rate is progressively increased, starting with a pressure of 20% of the expected breakdown pressure, and increasing the pressure by another 20% in the subsequent steps until the breakdown pressure is reached (Fig. 3b). The third testing procedure is based on the second test procedure with an added periodic pressure pulse on top of the progressively increasing pressure levels (Fig. 3c, details in Zimmermann et al. (2019)).

### 2.3.2 Hydraulic well test analysis

The evolution of permeability $k$ and transmissivity $kh$ is calculated from fall-off sequences (shut-in periods) of the declining pressure curves after each injection stage according to classical well test analysis (e.g. Horne, 1995). Hence, permeability is calculated taking into account the superposition principle and assuming infinite acting radial flow:

$$k = \frac{q\mu}{4\pi h \Delta p} \ln\left(1 + \frac{t_0}{\Delta t}\right), \tag{2}$$

where $h$ is the interval length, $q$ the flow rate, $\mu$ the dynamic viscosity of the injection fluid, $t_0$ the injection end time, and $\Delta t$
the shut-in time, differential pressure $\Delta p$ between initial pressure at shut-in and actual pressure.

The calculations (details in Zimmermann et al., 2019) are based on the slope of the declining pressure curves after shut-in and are performed for the last 100 seconds of each shut-in period for the conventional treatment with constant flow rates and for the last 40 seconds for the cyclic stages (Fig. 4). This ensures that stable and comparable radial flow conditions for each stage are obtained. The time differences are based on the shut-in time and are for the conventional treatments in the range of
300-700 s and for the cyclic stages in the range of 120-130 seconds.

### 2.3.3 Lugeon Tests

The Lugeon test is a pressure test carried out with a straddle packer in an isolated borehole interval (e.g. Lancaster-Jones, 1975, and references therein). Pressures and flow rates are recorded until quasi-steady state conditions are reached. Typical Lugeon tests consist of several pressure levels in a step rate way of increasing and decreasing pressure levels. In case of low permeable
rocks, often only single stage Lugeon tests are performed, since it might be difficult to obtain quasi-steady state conditions in a

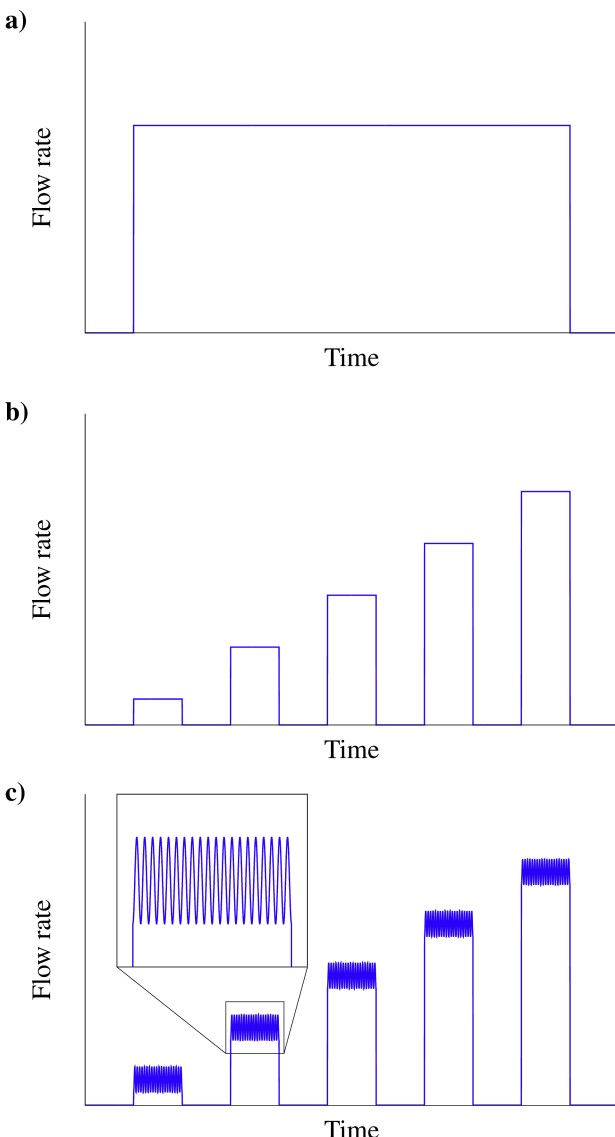

**Figure 3.** Schematics of the different testing procedures showing flow rates. a) conventional hydraulic fracturing procedure with constant flow. b) progressively increasing flow rate with a shut-in between cycles. The pressure is increased in steps of approximately 20% of the expected breakdown pressure estimated from the previous conventional test in the same rock type. c) progressively increasing flow rate with pressure pulses from a second pump system in addition to the progressively increasing pressure levels.

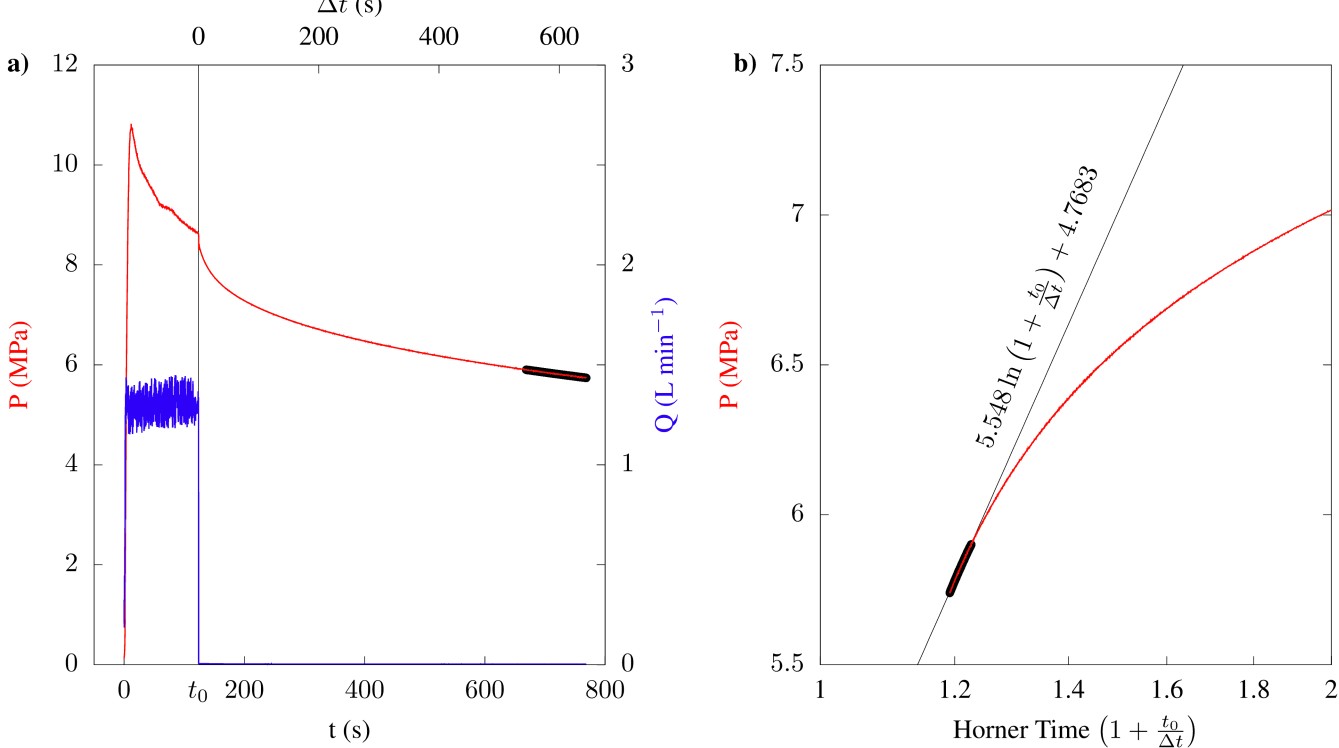

**Figure 4.** Determination of permeability a) Example of pressure and flow rate with time of the conventional fracturing procedure (Fig. 2a) from test HF1 Refrac 1 with constant flow rate (blue). Permeability is related to the slope of the pressure decline curve (red) after shut-in at time $t_0$ and is estimated from the last 100 seconds (thick, black line) of the shut-in period $\Delta t$. b) Same as a) with time represented as Horner time to estimate the slope (thin, black line) from the data highlighted by the thick, black line according to Eq. 2.

short time. Interval permeability is then obtained according to semi-empirical formulas by Hvorslev (1951) and Moye (1967) or the steady-state flow solution by Dagan (1978). A comparison of the aforementioned approaches with numerical solutions is given by Braester and Thunvik (1984).

### 2.3.4 Impression Packer Tests

We performed impression packer tests to obtain an imprint of the borehole wall to image the generated fracture traces after the hydraulic fracturing tests. The impression packer tool is able to measure the orientation of induced or stimulated fractures and consists of a single packer element with a soft rubber sleeve in conjunction with a magnetic single-shot orientation device. The packer preserves the imprint from the borehole wall and the fracture traces when it is pressurized to a pressure level higher than the fracture re-opening pressure for a duration of about 10 minutes.



**Table 1.** Borehole naming conventions and sensor installation with orientation (H - horizontal, V - vertical).

| Borehole label | Äspö HRL naming convention | Diameter (mm) | Sensors & orientation |
|---|---|---|---|
| F1 | KN0033B01 | 101.6 | Injection/fracturing |
| M1 | KN0021B01 | 75.8 | AE08, AE09, AE10 |
| M2 | KN0047B01 | 75.8 | AE03, AE04, AE05 |
| M3 | KN0048B01 | 75.8 | AE01, AE02 |
| M4 | K02016C01 | 76 | AE11, V |
| M5 | K02016F01 | 42 | ACC13, V |
| M6 | KN0040D01 | 76 | AE06, V |
| M7 | KN0040B01 | 42 | ACC16, H |
| M8 | KN0028D01 | 76 | AE07, V |
| M9 | KN0028B01 | 42 | ACC15, H |
| M10 | KN0018B01 | 42 | ACC14, H |

**Table 2.** Rock properties compiled from Staub et al. (2002, 2004); Nordlund et al. (1999); Stille and Olsson (1989, 1990). Percentage of rock types was inferred from tunnels KAS02, KA2511, KA2598 and is assumed to be representative for the whole Äspö rock volume.

| Rock type (naming convention of 2009) | Percentage of rock type | Young's Modulus $E$ (GPa) | Uniaxial compressive strength $\sigma_c$ (MPa) | Poisson's ratio $\nu$ | Density $\rho$ (kg m$^{-3}$) | Friction angle $\phi$ (°) | Tensile strength $\sigma_{ti}$ (MPa) |
|---|---|---|---|---|---|---|---|
| Äspö diorite | 34.9 | 73.0±2.9[1] 60.0 [2] | 214[3] 169 [4] | 0.27 | 2750 | 49 | 14.8 |
| Ävrö granodiorite | 53.9 | 62.0±0.50 | 182 | 0.24 | 2640 | 45 | 12.8 |
| fine-grained granite | 8.2 | 65.0±4.3 | 228 | 0.24 | 2640 | 45 | 15 |
| gabbroid-dioritoid | 3.0 | 52.5±17.4 | 115 | 0.22 | 2960 | 45 | 8 |

Images of fracture traces were obtained for all six hydraulic fracturing tests performed at Äspö HRL. The circumference of the packer is about 310 mm, the length of the pressurized interval is approximately 1 m. The images of all impression packer tests are available in Zimmermann et al. (2019, Appendix B).

## 3   Data description

The experimental setup with the central injection borehole F1 is surrounded by three inclined monitoring boreholes (Fig. 1).
Sensors are implemented in boreholes and adjacent tunnels at the depth level of 410 m in Äspö HRL. The summary and structure of all data published is summarized in Figure 5.





```
root
├── geological_data
│   ├── borehole_images..........................  Images of the injection borehole (F1) before    .jpg
│   │                                               (2015-05-18) and after (2015-06-17) the experi-
│   │                                               ments
│   └── core
│       ├── wet................................  Pictures of wet cores in core boxes (4 major bore-   .jpg
│       │                                         holes)
│       └── dry................................  Pictures of dry cores in core boxes (4 major bore-   .jpg
│                                                 holes)
├── hydraulic_data
│   ├── hf..................................  Time series of flow rate and pressure data during    .csv
│   │                                          the six experiments
│   └── lugeon..............................  Time series of flow rate and pressure data during 5  .csv
│                                              Lugeon tests
├── seismic data
│   ├── seismometer.........................  three-component data from 5 broadband seis-          .mseed
│   │                                          mometers installed in adjacent tunnels
│   ├── AE_triggered
│   │   ├── injection.......................  Triggered waveforms of 196 events (catalogue of   .mseed, .txt
│   │   │                                      Kwiatek et al., 2018), trigger reports, trigger setup
│   │   ├── pulse_transmission..............  Triggered waveforms of the pulse transmission     .mseed, .txt
│   │   │                                      tests, trigger times of pulses
│   │   ├── hammer ..........................  Triggered waveforms of hammer seismics, trigger   .mseed, .txt
│   │   │                                      times of hammer
│   │   └── catalog..........................  AE event catalogue from triggered records          .csv
│   │                                           (Kwiatek et al., 2018)
│   ├── AE_continuous
│   │   ├── HF1 ... HF6......................  Continuous waveforms recorded during the injec-    .mseed
│   │   │                                      tion experiment HF1 to HF6
│   │   ├── Lugeon ..........................  Continuous waveforms recorded during the Lu-       .mseed
│   │   │                                      geon tests
│   │   ├── noise............................  Continuous records of general noise                .mseed
│   │   ├── background.......................  Continuous records of pumping background noise     .mseed
│   │   │                                      and pumping simulations
│   │   ├── pulse_transmission..............  Continuous waveforms recorded during the pulse     .mseed
│   │   │                                      transmission tests
│   │   └── catalog..........................  AE event catalogue from continuous records         .csv
│   │                                           (Niemz et al., 2020)
│   ├── symphony.............................  Movie and audio files from modulated AE wave-    .mov, .wav
│   │                                          forms
│   └── reports..............................  Reports of commercial partners (GmuG,Mesy,         .pdf
│                                               ISATech)
```

**Figure 5.** Data structure of the online repository for the Äspö experiment.



**Table 3.** Fracture orientation from impression packer tests. Angle $\theta$ indicates fracture strike direction (East of North), angle $\beta$ indicates the dip direction (East of North) and angle $\alpha$ indicates the dip with respect to horizontal direction.

| Fracture trace | Strike $\theta$ (°) $0° < \theta \leq 180°$ | Dip azimuth $\beta$ (°) $0° < \beta \leq 360°$ | Dip $\alpha$ (°) $0° < \alpha \leq 90°$ |
|---|---|---|---|
| HF1 A | 153 | 243 | 74 |
| HF2 A | 118 | 208 | 60 |
| HF3 A | 51 | 141 | 47 |
| HF3 B | 116 | 206 | 84 |
| HF4 A | 177 | 087 | 85 |
| HF4 B | 83 | 353 | 63 |
| HF4 C | 147 | 237 | 89 |
| HF4 D | 129 | 219 | 77 |
| HF5 A | 68 | 158 | 49 |
| HF6 A | 127 | 217 | 80 |
| HF6 B | 67 | 157 | 54 |
| HF6 C | 145 | 55 | 71 |
| HF6 D | 121 | 211 | 62 |

### 3.1 Metadata

We provide geographical/location metadata of the Äspö experiments in comma-separated values (csv) files, accessible with any text editor or spreadsheet application. Additionally, we include a text-file containing the work protocol of the experiments (timeline). All location information is given in the ASPO96 coordinate system as described in section 2.1.1 and Eq. 1. The geographical data comprises 2D coordinates of the tunnel system at the 410-m level (as seen in Figure 1), borehole coordinates, and sensor locations. The four major boreholes F1, M1, M2, M3 are described by coordinates measured every meter during the drilling and the rock type at the given position. The other short boreholes (both horizontal and vertical) are described by a single coordinate due to their proximity of 0.5–1 m (Table 1).

The acoustic emission sensors (AE, GMuG MA BLw-7-70-75) and the accelerometers (ACC, Wilcoxon 736T.) were installed within the boreholes mentioned above (Table 1). ACC sensors were oriented horizontally, measuring in the direction of the borehole. Apart from the location, we provide the exact orientation of the AE sensors described by azimuth (clockwise from the y-axis seen from the access tunnel (TASN)) and inclination (clockwise from upward). The absolute instrument response function of the AE sensors is missing, and instead, AE sensor manufactures typically provide information about some aspects of the sensor response, e.g. dominant resonance frequencies or the excitation response, but this alone does not allow to obtain the full information on absolute ground motion (Plenkers et al., 2022). The frequency response of the AE sensors deployed at Äspö HRL is expected to be flat between 10 Hz and 10 kHz (see GmuG report in the data archive). Broadband seismometers (BB, Trillium Compact 120s) were installed in adjacent tunnels with unknown instrument orientation (Niemz et al., 2021).

The orientation of the sensors can be estimated using teleseismic or regional earthquakes (Petersen et al., 2019; Niemz et al., 2021). We provide a text file (.paz) with poles and zeros (e.g., IRIS, https://ds.iris.edu/ds/nodes/dmc/data/formats/resp/) for the response of the instrument and the used digitizer.

### 3.2 Geological data

We provide photos of the cores in core boxes in wet and dry conditions for each of the four major boreholes (Fig. 2). For the injection borehole (F1) we provide additional imagery of the borehole wall taken with a borehole camera system . The rock types can be matched with the information given in the borehole metadata (precision 1 meter). Average rock mechanical properties are listed in Tab. 2.

### 3.3 Hydraulic data

Hydraulic data is provided as time series of flow rate and interval pressure as csv-files. For the experiments HF1 to HF3, the files also include the packer pressure. In Table 3, the orientations of the fracture traces from the impression packer results are summarized. The hydraulic data defines the start and stop of injections, shut-in phases, and bleed-off phases. The relative timing difference between the AE waveforms and the hydraulic time series is expected to be smaller than 1 second for HF1 to HF3. For HF4 to HF6, the relative timing could only be reconstructed by comparing the timing between the hydraulic data, the AE activity and the pumping signal in the continuous AE records.

### 3.4 Seismological data

We provide three different data sets with seismological waveform data: continuous broadband seismometer waveforms (Section 3.4.1); triggered, high-frequency waveforms of the AE sensors and the accelerometers (Section 3.4.2); and continuous, high-frequency waveforms of the AE sensors and the accelerometers (Section 3.4.3). The waveforms are provided in miniSEED format. MiniSEED is a standard seismological format, that can be opened with any seismological tool box, such as Obspy (available at obspy.org) or Pyrocko (available at pyrocko.org). The sampling rate of the seismic traces differs depending on the instrument type. The seismometer data is sampled at 100 Hz, while ACC and AE data is sampled at 500 kHz. The high-precision sampling of 2 milliseconds of the AE and ACC sensors is handled correctly by the two applications mentioned before, other applications were not tested. Using 64-bit floating point numbers can lead to rounding errors in the millisecond range. Using 128bit floats is recommended if rounding errors become a problem in some applications. Waveform data is provided as station-wise archives for the seismometers and the continuous AE waveforms. The triggered records are combined in a single archive.

#### 3.4.1 Broadband seismometers

The 3-component data recorded by the five broadband seismometers (Trillium Compact 120s) covers approx. two weeks along with the Äspö experiment. The output unit is counts (see metadata for instrument response information).





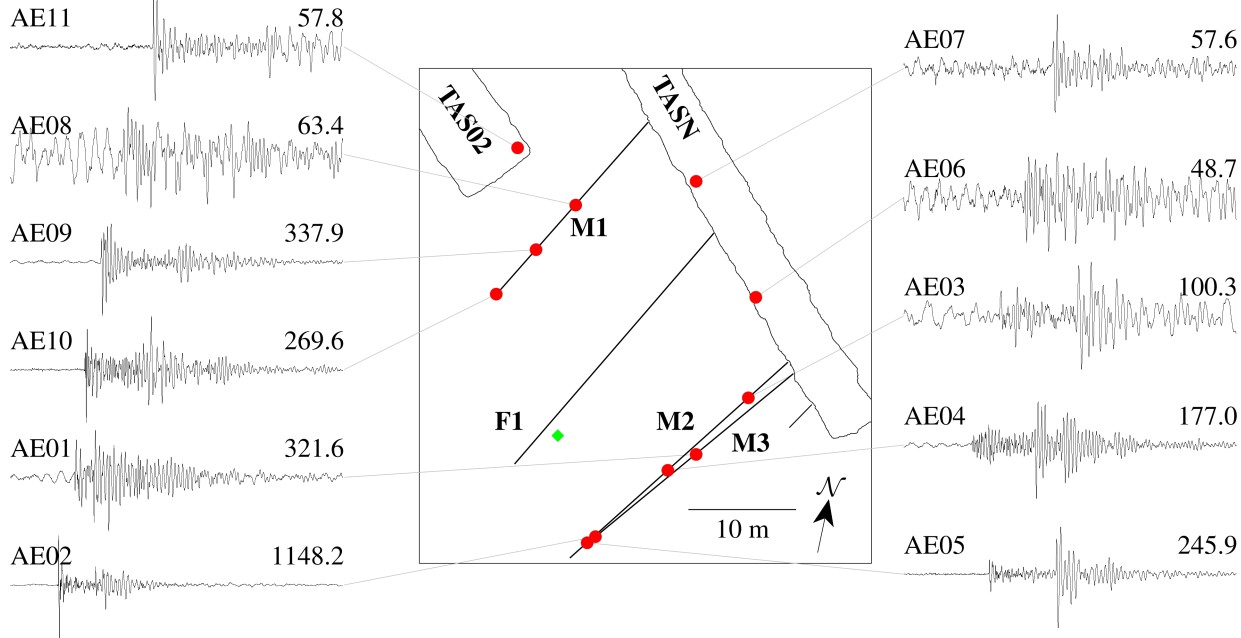

**Figure 6.** Waveforms of acoustic emission sensors for the first event recorded in trigger mode during test HF1 Refrac 1 (green diamond). The sensors corresponding to the waveforms are shown as red dots in map view. The three long monitoring holes M1-M3 are indicated as black lines. Sensors in short boreholes (AE06, AE07, AE08) were installed in the tunnel ceilings. Note, that holes M2 and M3 have different inclinations (compare with Fig. 1), and thus the vertical coordinates of the sensors differ. This explains the arrival time difference for AE02 and AE05, despite similar (x,y)-coordinates. Waveforms are normalized with scale factors given at the end of each trace. Time window length of the records is 10.3 ms.

### 3.4.2 Acoustic Emission and Accelerometer data: triggered records

The triggered recording system was active during the whole period of the experiments with 15 single-component sensors (11 AE and 4 ACC sensors). When the given threshold is reached, the system saves 32.768 ms of data from each sensor (AE and ACC). The low trigger level resulted in many false detections originating from noise signals. Therefore, we do not include all triggered records but provide only a subset of manually double-checked data from three phases of the experiments. If needed, the triggering can be simulated using the continuous records and the setup of the triggering system, which is provided with the

triggered waveforms of the injection experiments. One example of waveforms from HF1 Refrac 1 is shown in Fig. 6.

1. **Hammer seismics** This data set comprises the hammer tests on June 1, June 3 (before the stimulations), and June 15 (after the stimulations).

2. **Pulse transmission** The data comprises waveforms of multiple pulse transmission tests on June 2 and a single test on June 1 (before the stimulations) and on June 15 (after the stimulations). The signal on ACC13 does represent acceleration

data during the pulse transmission test. During pulse transmission, this channel was used to record the pulse signal.

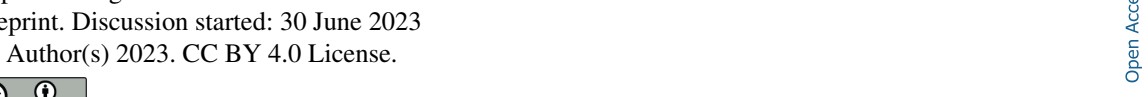


3. **Injection experiments HF1-HF6** The data contains the waveforms of the preliminary AE catalogue (196 events) from Kwiatek et al. (2018). Additionally, we provide the trigger reports and a csv-file with event matches between the triggered catalogue AE catalogues of Kwiatek et al. (2018), and the continuous catalogue of Niemz et al. (2020).

### 3.4.3 Continuous records

Due to the high sampling rate of the ACC and AE sensors, the continuous recording system was only activated during tests and experiments and for noise recordings on the 15 channels. The continuous and the triggered recording system are not synchronized. Timing can differ up to a few seconds, but waveforms are identical. For information on the activities during the continuous recordings we also refer to the work protocol within the metadata. We provide waveforms for the following experiments and tests:

1. **Injection experiments HF1- HF6** The continuous waveforms begin a few minutes before the injection and end several minutes (up to 30 min possible) after the bleed-off phase.

2. **Lugeon Test** The data comprises continuous waveforms during the Lugeon test.

3. **Pulse transmission** The data comprises the continuous waveforms of the pulse transmission tests on 2 June. As for the triggered data, ACC13 respresent the pulse signal itself.

4. **Ventilation noise** Noise was recorded with active tunnel ventilation (4 June 2015, around 14:39) and inactive tunnel ventilation (4 June 2015, around 14:55). Tunnel ventilation was always operating during the experiments and this is the only recording without ventilation noise. A third noise measurement was done on 10 June, around 11:29. See also the work protocol within the metadata.

5. **Background noise** Records of pumping equipment test can help to evaluate the background noise conditions during the
injection.

We provide continuous waveforms from a test of the hydraulic hammer only (5 June, around 14:05), a test of the pump only (around 14:34) and a test of both (around 14:58). Additionally, there are continuous records from a repeated simulation of the fatigue fracturing technique using the hydraulic hammer on 10 June, around 12:09. See also the work protocol.

### 3.5 Hydraulic fracture "symphony"

Here, we provide audio files produced from the waveforms of the AE sensors. The AE sensors cover completely the human audible range but are of course much more sensitive than human hearing. When the waveform data is converted to sound files, various events during the tests become audible. For the actual fracking, the onset of the first refrac of first test (HF1-RF1) is made audible. In the real-time version, the injection pump is noticeable as metallic hum. When the injection process is continuous, more and more crackling sounds (like fire or plastic wrap) appear over time, which are tiny cracks in the rock
mass, continuously fracturing. A second version of HF1-RF1 replays the sound at 1/25th the original speed. The pump is now





a clear monotonous signal and the fracturing process sounds like random knocking sounds from some distance away. The background noise is mostly due to the ventilation system in the access tunnels. Another example is a single hammer strike recorded at various sensors. As the strikes are strongest signal recorded during the experiment, their corresponding sound is clear on almost all sensors. The sledgehammer sound file is played at 1/100th of the original speed. Another constant acoustic
source is water dripping into the boreholes through preexisting fractures, connected to the Baltic Sea above Äspö HRL. From various cracks water dripped into the boreholes (water sound file), where it accumulated and then spilled into the access tunnels (water outflow movie).

### 3.6    Reports of commercial partners

We provide the reports of the commercial partners of the Äspö experiments containing additional information on the timeline
of the experiments and the first results.

### 4    Data quality

Data quality follows established standards, however, given the underground location of the experiment, absolute timing issues did arise. Relative timing, especially for the borehole monitoring network on the other hand is of high precision.

Packer and interval pressure were measured in borehole F1 with high precision electric pressure transducers (KELLER, type
PAA-33X, 0 - 40 MPa). The pressure values and injection rate (RCI flow-meter, type QPT04) were recorded by a digital data acquisition system (Solex-perts SCI-A, 16 channels, 16 bit resolution, sampling rate: 5 Hz). As part of the quality assurance, the pressure transducers and the flow-meter were calibrated prior to in-situ testing.

The output unit of the broadband records is in counts and can be converted to velocity using the instrument response information provided with this data set. The data set contains a few gaps of up to 10 min (mostly one or two gaps over a
range of 3 weeks). These gaps may arise from temporary instrument malfunctions. The location of the seismometers within the tunnel should be considered as approximate w.r.t. locations in the tunnels, since there is inherently no GPS signal within the mine. Consequently, the seismometer clocks were also not automatically corrected via GPS, resulting in potential clock drifting. Additionally, the seismometers were not rigorously synchronized with the AE and ACC instruments and time offsets of a few seconds between AE data and broadband data are possible. The seismometers were oriented arbitrarily (see channel
names 1 and 2 instead of N and E) with one horizontal component approximately parallel to the closest tunnel wall.

The AE sensors used in this experiment were not calibrated. The absolute sensor response in terms of velocity or acceleration is unknown and influenced by the incidence-angle-dependent response of the piezoelectric sensor to incoming waves and the varying coupling of the sensors to the borehole wall. The continuous recording of AE/ACC data was only active during times of injection experiments. Due to a miscommunication, there is only partial data for the initial fracturing stage of the injection
experiment HF2. The absolute timing between the data of different injection experiments or stages if the experiment was split up, is only correct to 1 or 2 seconds, since the system was started manually each time. In contrast, the relative timing between all AE and ACC sensors can be considered to be as precise as the sampling interval, as it was synchronized via the recording





system. The high-precision sampling of 2 milliseconds of the AE and ACC sensors is handled correctly by obspy (available at obspy.org) or Pyrocko (available at pyrocko.org). Other applications were not tested. The record time data should be treated as 310 128-bit floats. Otherwise, rounding errors may become a problem.

The triggered waveforms represent a subset of the continuous data set, with 32.768 samples around a manually revised, in-situ detected event. The waveforms of the continuous and triggered data sets are identical, but the record times between them may vary by several seconds. The different data sets can be matched using waveform cross-correlations.

## 5   Data usage in previous articles

In an overview, Zang et al. (2017) describe the experimental setup of sensors and injection schemes in the geologic context of Äspö HRL. A preliminary catalogue of the strongest and most reliably recorded events was compiled (196 events) with origin times and hypocenters of the acoustic emissions in six hydraulic in-situ tests with 29 fracturing stages. In two tests, acoustic emission, hydraulic pressure and flow-rate time charts are analyzed, and discussed in light of the stress state and the fracture orientations from impression packer results.

López-Comino et al. (2017) apply a robust, accurate and automated detection and location algorithm of acoustic emissions to characterize the nucleation and growth process of hydraulic fractures. Here, full waveform recordings monitored during one test were selected. Waveform stacking and coherence analysis techniques are applied, using large data sets with 1 MHz sampling from the continuous water injection test. A catalogue of 4000 acoustic events is generated with a high Gutenberg-Richter $b$-value of 2.4. Fracture growth is mapped by the spatiotemporal evolution of acoustic emission locations revealing 325   upwards migration from -414 m to -404 m depth.

Zang, Stephansson & Zimmermann (2017) introduced the concept of fatigue hydraulic fracturing as a possible explanation for the above mentioned observations. This concept is based on alternating phase of pressurization and depressurization allowing crack-tip stresses to periodically relax. Treating fracture walls with a hydraulic hammer moves rock chips to the fracture tip as described in the Kiel process (Kiel, 1977). This makes the fracture process zone to become larger in the fa-330   tigue test as compared to the conventional treatment. The multiple-pump, variable-flow rate approach allows for efficient rock fragmentation.

Kwiatek et al. (2018) analyzed triggered recordings of eleven acoustic emission sensors and determined moment magnitudes ranging from -4 to -3.5 using acoustic emission and accelerometer data from the preliminary catalog. Events have been relocated with the double-difference technique and investigated in combination with the source parameters in the context of 335   stimulation parameters. Migration of the events away and toward the injection intervals was observed in direct correlation with changes in hydraulic energy. Total radiated seismic energy is identified to be very low with respect to the hydraulic energy and correlates well with the hydraulic-energy rate. Source parameter analysis signify the reactivation of preexisting rock defects.

Zimmermann et al. (2019) used hydraulic test data to compute permeability for the six hydraulic tests with 29 (re)fracturing stages. The evolution of permeability with time is compared to the number of triggered acoustic emission recordings from





Kwiatek et al. (2018). Compared to conventional injection tests with constant flow rate and monotonic increase of the fracturing pressure, the cyclic injection leads to a lower activity of acoustic emissions.

Zang et al. (2019) compare laboratory cyclic injection tests in granitic rock with mine-scale data from the Äspö HRL in-situ tests. General findings independent of scale, are: (a) a lower breakdown pressure in fatigue testing, (b) a reduction in the magnitude of the largest induced seismic event by cyclic injection, (c) a wider process zone in the cyclic fracture patterns, (d)
an increase in permeability during cyclic injection, although this increase is less compared to that in continuous injection.

Stephansson et al. (2019) tested cores obtained from the injection borehole used for the in-situ tests performed at Äspö HRL. The testing procedure is designed with linear pressure increase for both conventional and pulse fracturing. For each pair of samples of present rock types, testing starts with continuous hydraulic fracturing until linear breakdown. The loading steps of the pulse fracturing test are given as percentages (25%, 50% and 75%) of the linear breakdown pressure. The pulse
frequency for each loading step is set to 1 Hz. The axial pressure is set to 8.5 MPa and gives a stress ratio between confinement and axial stress of 1.4. The applied stress ratio value is adapted to in situ values (maximum horizontal 22 MPa, vertical 12 MPa and minimum horizontal stress 11 MPa at a depth of 410 m).

Niemz et al. (2020) developed a semi-automated work flow with full-waveform-based detection, classification and localization procedures for six hydraulic in-situ tests to extract and characterize the intense activity of induced, high-frequency
acoustic emissions from continuous recordings. The approach extends the AE catalogue from 196 events of the preliminary catalogue to more than 19,600 located acoustic emissions. While the conventional tests lead to hypocenters clustered in planar regions, indicating the generation of a single main fracture plane, the cyclic progressive injection scheme results in a more diffuse hypocenter distribution, indicating the activation of a more complex fracture network. In terms of hydraulic energy, the cyclic progressive scheme is characterized by a lower rate of seismicity, lower maximum magnitudes and larger $b$-values; the
last implying an increased number of small events relative to larger ones. The magnitude distributions of the catalogues by Kwiatek et al. (2018) with moment magnitudes and Niemz et al. (2020) with acoustic-emission magnitudes are shown in Fig. 7.

Zang et al. (2021) propose a new approach that optimizes the trade-off between the unavoidable radiated seismicity during fracture propagation and the inserted hydraulic energy during fluid injection by using cyclic- and pulse-pumping schemes.
Their concept aims at the ability to control induced seismicity in energy technologies such as geothermal heat and shale gas, and conclusively improving the safety by reducing the seismic hazard of reservoirs. They use data from laboratory-, mine-scale, and field-scale injection experiments performed in granitic rock and observe that both the seismic energy and the permeability-enhancement process strongly depend on the injection style and rock type.

Niemz et al. (2021) analyze tilt signals that appear as long-period transients on two broadband seismometers installed in
proximity to newly formed, meter-scale hydraulic fractures. The tracking of increased permeability and the fracturing extent is often based on the distribution of accompanying microseismic events within the stimulated rock volume, but it is debated whether microseismic activity adequately depicts fracture formation. The analysis combines a catalogue of previously analyzed acoustic emissions, indirectly mapping the fractures, with tilt signals providing an independent and direct insight into rock

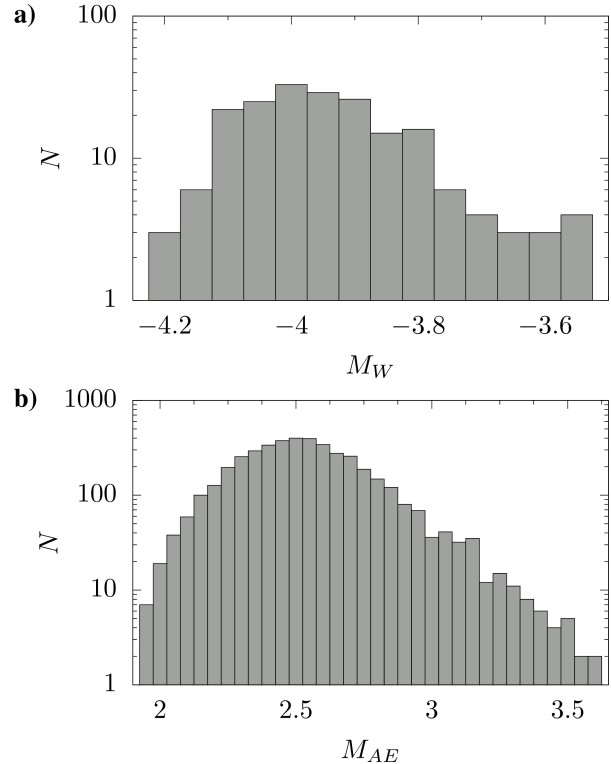

**Figure 7.** Magnitude-frequency plots of a) the catalogue of Kwiatek et al. (2018) and b) the catalogue of Niemz et al. (2020).

deformation. The analysis allows to identify different phases of the fracturing process including the (re)opening, growth, and

after-growth of fractures.

López-Comino et al. (2021) used Äspö HRL strong acoustic events to estimate rupture directivity of the hydraulic fractures. High-quality waveforms recorded from the largest, decimeter-scale acoustic emissions during the in-situ experiment allow to resolve the apparent durations observed at each sensor to analyze 3D-directivity effects. Unilateral and (asymmetric) bilateral ruptures are then characterized by the introduction of a parameter $\kappa$, representing the angle between the directivity vector and

the station vector. While the cloud of acoustic emission activity indicates the planes of the hydraulic fractures, the resolved directivity vectors show off-plane orientations, indicating that rupture planes of micro-fractures on a scale of centimeters have different geometries. The results reveal a general alignment of the rupture directivity with the orientation of the minimum horizontal stress, implying that not only the slip direction but also the fracture growth produced by the fluid injections is controlled by the near-field, local stress conditions.

Zang et al. (2019) and Zhuang et al. (2019) analyzed the energy budget during hydraulic fracturing with the goal to control the energy partition in the fracture growth process. The hydraulic energy in the underground tests were estimated to be 0.2 MJ, while hydraulic energy in the cyclic laboratory tests varies between 10 J and 10 kJ, increasing the number of cycles from one to 839. The reported findings can improve the efficiency of heat production in granitic rock, or shale gas production with



an environment-friendly approach which (a) reduces the fracture breakdown pressure of high-strength rocks (smaller pumps
required), (b) reduces induced seismicity (expanding the operating time of the project), and (c) generates a larger stimulated
reservoir volume with natural proppants from the fracture walls by rock hydraulic fatigue, labeled "soft stimulation" or "fatigue
hydraulic fracturing" technique.

There are many data sets which call for further analysis (Fig. 5, e.g. the ultrasonic pulse transmission data and the noise data
before and after in-situ experiments were not investigated, yet.

*Data availability.* The data will be published via GFZ Data Services (Zang et al., 2023) after the finalized review of the article. In the
meantime, the data are available via this temporary link: `https://dataservices.gfz-potsdam.de/panmetaworks/review/`
`4a502c148420d56db09f93239232e27cb5a91054fcfda62f55b138777a45cd05/`. The data repository is structured in direc-
tories (Figure 5) and available as compressed archive using gzip (file format tar.gz, Windows and Mac users may need additional software to
open the archive, e.g. winzip or 7zip).

*Author contributions.* Arno Zang, Peter Niemz and Günter Zimmermann conceptualized this study. Arno Zang wrote the first draft of this
manuscript and was principal investigator of the 2015 mine-scale hydraulic stimulation experiments at Äspö Hard Rock Laboratory. Arno
Zang, Claus Milkereit, Sebastian von Specht, Katrin Plenkers, Gerd Klee were members of the team collecting all data underground. Peter
Niemz structured the database and prepared data files and formats for data hosting. Sebastian von Specht edited the draft and designed Fig.
2-7, Fig. 1 by Peter Niemz. All authors read, revised and approved the final version of the manuscript.

*Competing interests.* All authors declare no competing interests.

*Disclaimer.* The authors The data described in this article and accessible at `https://doi.org/10.5880/GFZ.2.6.2023.004` are
provided on "as is" and "as available" basis without warranty of any kind.

*Acknowledgements.* The in-situ experiment at Äspö Hard Rock Laboratory (HRL) was supported by the GFZ German Research Center for
Geosciences (75%), the KIT Karlsruhe Institute of Technology (15%), and the Nova Center for University Studies, Research and Develop-
ment Oskarshamn (10%). We thank Hana Semíková and Ondřej Vaněček at ISATech s.r.o. (Prague, Czech Republic) for hydraulic stimulation
of test HF4 - HF6. An additional in-kind contribution of the Swedish Nuclear Fuel and Waste Management Co (SKB) for using the Äspö
HRL as test site for geothermal research is highly appreciated. This work was supported by funding received from the European Union's
Horizon 2020 research and innovation programme, grant agreement No. 691728 (DESTRESS).



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
