# Peer review of "Comprehensive data set of in-situ hydraulic stimulation experiments for geothermal purposes at the Äspö Hard Rock Laboratory (Sweden)"

_Earth System Science Data, 2023_

## Author Comment (AC1)

Reviewer comments in normal font, our responses in italics.

Very nice paper. I have a few comments.

Abstract - State data location data will be archived

*The status of the data location and availability will be updated once the paper is finalized (according to the journal policy).*

Line 39 Change "2018+ Homestake Mine underground tests at Saniford Facility in North Dakota," to "

2018+ EGS Collab Project underground tests at Sanford Underground Research Facility (SURF) in South Dakota,"

*Modified.*

Line 73 Explain the 29 fracture stages. To me a fracture stage is a location, but there are apparently 6 locations.

*We specified this further: Hydraulic tests were performed at six injection intervals (test intervals). In each hydraulic test interval, several fracturing and re-fracturing stages were performed. In total, 29 treatments (stages) of the granitic rock mass at six locations (test intervals) in the horizontal injection borehole were performed underground.*

Line 102 "long" should be quantified. These long boreholes are short to most people.

*"short" and "long" have been quantified in the text and a description of the "short" boreholes has been added.*

Section 2.3.1 State the intervals for these tests.

*We refer to Zang et al. (2017) for details on the test intervals and injection schemes used.*

Line 169 Change "permeable" to "permeability".

*Modified.*

Table 1. "M4" to "M10". Verify that these are described in the paper.

*Fully agree. These were indeed missing in the text. We added a short description in the text and included the borehole labels also in Fig. 1.*

Line 202 The location or reference for the "data archive" needs to be stated.

*We have no control on this. As stated above, this will updated by the journal once the paper is finalized.*

Figure 6. Define the numbers above each trace.

*A description is given in the figure caption "Waveforms are normalized with scale factors (in mV) labeled at the end of each trace." We added the unit of the scale factor "(in mV)".*

Line 253 State the location of the metadata.

*As stated above, this will updated by the journal once the paper is finalized.*

---

## Author Comment (AC2)

Reviewer comments in normal font, our responses in italics.

Very interesting paper, written in a clear language. In the framework of the renewable energy sources and induced seismicity, it is a work that addresses current issues by presenting interesting food for thought and a complete and useful data set.

*Thank you for overall positive feedback.*

Few observations:

- Line 59: change "Zang et al. (2017)" in "Zang et al (2017a)" as reported in the bibliography. As well as for the other 2017 Zang article: "Zang, Stephansson & Zimmermann (2017)" in "Zang et al (2017b)"

  *Modified accordingly.*

- Line 73: Could you better explain in what the 29 fracture stages consist of compared to the six hydraulic tests?

  *We specified this further: Hydraulic tests were performed at six injection intervals (test intervals). In each hydraulic test interval, several fracturing and re-fracturing stages were performed. In total, 29 treatments (stages) of the granitic rock mass at six locations (test intervals) in the horizontal injection borehole were performed underground.*

- Line 88: I suggest replacing "Geology" with "Geological methods" in line with the other sections (Borehole geophysical methods; Hydraulic methods)

  *Modified accordingly.*

- Line 102: Highlight the location of TASA and TAS02 tunnels in relation to TASN tunnel in figure 1 (if it is possible)

  *Labels for tunnels and also short boreholes have been added.*

- Line 136: Add a reference to the technique used to estimate P and S velocity starting from the ultrasonic pulse transmission data

  *We added the reference Zang et al. (2017a), which discusses in more detail the velocity estimation.*

- Figure 2: "HF1" explain in the text what it refers to. I think it is referred to the six hydraulic tests but it is not explicitly explained; as well as for the following HF? and HF? Refrac codes.

*Labels HF1 to HF6 indicate the fracturing tests. For each test, e.g. HF1, labels HF1-F, HF1-RF1, HF1-RF2 etc. indicate the initial fracturing and multiple re-fracturing stages, respectively. This has been added to the text.*

- Figure 4: Speaking of "c*onventional fracturing procedure*", why are you referring to figure 2a ? Is it not figure 3a?

  *Fully agree. Modified accordingly.*

- Lines 161-163: "The calculations (details in Zimmermann et al., 2019) are based on the slope of the declining pressure curves after shut-in and are performed for the last 100 seconds of each shut-in period for the conventional treatment with constant flow rates and for the last 40 seconds for the cyclic stages (Fig. 4)". Does the figure refer only to the case of constant flow rates, right?

  *Yes, it is correct that Fig.4 refers to the case of a constant flow rate.*

- Table 1: M4 e M10 - what are their locations with respect to the area and the injection borehole?

  *Fully agree. These were indeed missing in the text. We added a short description in the text and included the borehole labels also in Fig. 1.*

- Figure 6 : Is the number above each trace the scaling factor?

  *Yes, as stated in the figure caption: "Waveforms are normalized with scale factors (in mV) given   at the end of each trace." We added also the unit of the scale factor to the caption (mV).*

- Line 193: what do you mean by short boreholes? You only give the information about injection borehole length. What about the others(M1-M10)? I see that some of these informations are contained in metadata but you don't explicitly mentioned that in the text.

  *Fully agree. These were indeed missing in the text. We added a short description in the text and included the borehole labels also in Fig. 1.*

---

## Author Response (AR2)

Response to Editor

Dear Sebastian von Specht and co-authors
Considering your responses to the reviewers and the revised version of the manuscript, your paper is almost ready to be accepted for publication.

However, as pointed out by Reviewer #1 and according to ESSD Data Policy and Submission Guidelines, the data must have a functional DOI and citation and should be available under a non-restrictive license such as CC0 or CC BY.
As required by the Submission Guidelines, I encourage you to add a section entitled "Data availability" (either before or after Section 4 "Data Quality"), stating that your data, with their citation, are available via the GFZ Data Services at https://doi.org/10.5880/GFZ.2.6.2023.004, adding if possible the distribution conditions.
The provided DOI is not working at the moment, please contact the GFZ repository to register it.

If you have any questions, please do not hesitate contacting me or the ESSD editorial office.

Best regards,
Andrea Rovida
* * *
*As requested, we moved the data availability section after the data quality section, added the now available data link under https://doi.org/10.5880/GFZ.2.6.2023.004 (tested today), and stated the license for the data as CC BY (also stated on the data web website).*